# Enhanced Assessment of Cross-Reactive Antigenic Determinants within the Spike Protein

**DOI:** 10.3390/ijms25158180

**Published:** 2024-07-26

**Authors:** Guilherme C. Lechuga, Jairo R. Temerozo, Paloma Napoleão-Pêgo, João P. R. S. Carvalho, Larissa R. Gomes, Dumith Chequer Bou-Habib, Carlos M. Morel, David W. Provance, Thiago M. L. Souza, Salvatore G. De-Simone

**Affiliations:** 1Center for Technological Development in Health (CDTS), National Institute of Science and Technology for Innovation in Neglected Population Diseases (INCT-IDPN), Oswald Cruz Foundation (FIOCRUZ), Rio de Janeiro 21040-900, Brazil; gclechuga@gmail.com (G.C.L.); pegopn@gmail.com (P.N.-P.); larissagomesr@gmail.com (L.R.G.); carlos.morel@fiocruz.br (C.M.M.); tiago.moreno@fiocruz.br (T.M.L.S.); 2Cellular Ultrastructure Laboratory, Oswaldo Cruz Institute, Oswaldo Cruz Foundation, Rio de Janeiro 21040-900, Brazil; 3Laboratory on Thymus Research, Oswaldo Cruz Institute, Oswaldo Cruz Foundation, Rio de Janeiro 21040-900, Brazil; jairo.jrt@gmail.com (J.R.T.); dumith.chequer@gmail.com (D.C.B.-H.); 4National Institute for Science and Technology on Neuroimmunomodulation, Oswaldo Cruz Institute, Oswaldo Cruz Foundation, Rio de Janeiro 21040-900, Brazil; 5Graduate Program in Science and Biotechnology, Department of Molecular and Cellular Biology, Biology Institute, Fluminense Federal University, Niterói 24220-900, Brazil; 6Laboratory of Immunopharmacology, Oswaldo Cruz Institute, Oswaldo Cruz Foundation, Rio de Janeiro 21040-900, Brazil; 7Epidemiology and Molecular Systematics Laboratory, Oswaldo Cruz Institute, Oswaldo Cruz Foundation, Rio de Janeiro 21040-900, Brazil

**Keywords:** antibody-mediated enhancement, B-cell epitope, COVID-19, Dengue virus, SARS-CoV-2 variants, spike protein, vaccination

## Abstract

Despite successful vaccination efforts, the emergence of new SARS-CoV-2 variants poses ongoing challenges to control COVID-19. Understanding humoral responses regarding SARS-CoV-2 infections and their impact is crucial for developing future vaccines that are effective worldwide. Here, we identified 41 immunodominant linear B-cell epitopes in its spike glycoprotein with an SPOT synthesis peptide array probed with a pool of serum from hospitalized COVID-19 patients. The bioinformatics showed a restricted set of epitopes unique to SARS-CoV-2 compared to other coronavirus family members. Potential crosstalk was also detected with Dengue virus (DENV), which was confirmed by screening individuals infected with DENV before the COVID-19 pandemic in a commercial ELISA for anti-SARS-CoV-2 antibodies. A high-resolution evaluation of antibody reactivity against peptides representing epitopes in the spike protein identified ten sequences in the NTD, RBD, and S2 domains. Functionally, antibody-dependent enhancement (ADE) in SARS-CoV-2 infections of monocytes was observed in vitro with pre-pandemic Dengue-positive sera. A significant increase in viral load was measured compared to that of the controls, with no detectable neutralization or considerable cell death, suggesting its role in viral entry. Cross-reactivity against peptides from spike proteins was observed for the pre-pandemic sera. This study highlights the importance of identifying specific epitopes generated during the humoral response to a pathogenic infection to understand the potential interplay of previous and future infections on diseases and their impact on vaccinations and immunodiagnostics.

## 1. Introduction

The COVID-19 pandemic has imposed a high global burden of disease, with millions of lives lost and a tremendous economic cost. SARS-CoV-2 continues to adapt, and despite the rapid development of vaccines and successful vaccination strategies, several variants of concern have emerged. Most recently, the Omicron XBB.1.5. variant has gained attention, but other variants have been reported as well, including B.1.1.7 (Alpha), B.1.351 (Beta), P.1 (Gamma), and B.1.617.2 (Delta) [1]. The variants spread dynamically, and the dominant Omicron variants (VoCs) are the most important variants. Mutations especially in the spike RBD have displayed greater levels of infectivity and transmissibility [2]. Also, these variants have demonstrated a resistance to neutralization and escape neutralizing antibodies (nAbs) induced by vaccinations or prior infections [3,4]. These outcomes have led to the development of an updated vaccine to include mutations in the Omicron variant

The study of humoral response to SARS-CoV-2 is essential to mapping population and individual responses to viral infections and vaccinations. The antibody recognition of linear B-cell epitopes varies among individuals [5,6], but an immunodominant antibody analysis is important for serodiagnosis and prognosis [7].

Cross-reactivity is a major concern in developing diagnostic tests, and SARS-CoV-2 serological assays mostly use the glycoprotein spike (S) or N (nucleoprotein) full-length proteins or subdomains. Although this strategy increases the tests’ sensibility, it also increases the chance of false-positive results. Despite the low similarity of SARS-CoV-2 proteins to other human coronaviruses (HCoVs), some conserved regions can present cross-reactivity [4]. In addition, there is evidence that the response to SARS-CoV-2 infections appears to be shaped by previous HCoV exposure, which has the potential to lead to broadly neutralizing responses [4]. Interestingly, cross-reactivity with other endemic viruses has been uncovered, including with Dengue virus (DENV) and other respiratory viruses [4,7,8].

Antibodies against the SARS-CoV-2 spike protein and its receptor-binding domain (S1-RBD) were significantly increased in DENV-infected patients compared to normal controls [8]. In addition, anti-S1-RBD IgG antibodies purified from S1-RBD hyperimmune rabbit sera could cross-react with the DENV envelope protein (E) and non-structural protein 1 (NS1) [8]. Functionally, in vitro assays demonstrated that DENV infection and DENV NS1-induced endothelial hyperpermeability were inhibited in the presence of anti-S1-RBD IgG, and the passive transfer of anti-S1-RBD IgG induced some protection against DENV infection in mice [8]. Also, an in vitro analysis using COVID-19 patient sera showed neutralizing activity against Dengue infections [8].

Cross-reactive antibodies represent a double-edged sword; some can induce neutralization, but others can drive antibody-dependent enhancement (ADE). ADE is a phenomenon in which antibodies increase viral infection, as observed with DENV, yellow fever virus (YFV), West Nile virus (WNV), Ebola virus, influenza A virus [9,10,11], human immunodeficiency virus (HIV) [12,13,14], measles, and respiratory syncytial virus (RSV), but the exact mechanism is still uncertain [15].

Among other examples of ADE, secondary infection with a Dengue virus of the heterologous serotype has been associated with immunopathologic vascular leakage and hemorrhagic syndrome, Dengue hemorrhagic fever/Dengue shock syndrome (DHF/DSS) [16].

Virus–antibody complexes bind to Fc receptors (FcRs), expressed on immune cells, by the antibody’s fragment crystallizable (Fc) portion. Human mAbs against SARS-CoV-2 spike protein were found to enhance the viral infection in vitro by the FcγR-mediated pathway [17]. Another study using serum samples of acute and convalescent COVID-19 patients demonstrated ADE by FcγR-mediated and C1q-mediated pathways [18,19,20,21].

ADE also can induce enhanced immune activation [9]. SARS-CoV-2 ADE has been under debate since the beginning of the pandemic, especially due to concerns related to vaccination and the emergence of new variants. Although in vitro evidence supports a potential risk, no conclusive data have been reported that ADE is related to disease severity [22,23]. ADE is a rare event that requires many conditions associated with antibodies, viruses, and hosts [24,25]. Some events involved with antibody responses to SARS-CoV-2 are still unclear, such as the duration of humoral response and the pre-formed antibody repertoire due to previous infections. Each individual has a collection of memory B-cells and antibodies that can drive the immune response against SARS-CoV-2.

In this study, we mapped the immunodominant IgG linear B-cell epitopes and evaluated their cross-reactivity profile against pre-pandemic Dengue-positive serum samples. Several epitopes identified in the SARS-CoV-2 spike protein showed a similarity in terms of residue to DENV proteins, and we hypothesized that pre-formed DENV antibodies could interact with the SARS-CoV-2 spike protein. The data suggest that some sites in spike proteins cross-react with DENV pre-pandemic sera, and it could potentiate SARS-CoV-2 infections of monocytes through ADE in vitro. Many other studies have evaluated the cross-reactivity of coronaviruses and SARS-CoV-2 [26,27,28,29,30,31,32], and it is broadly distributed across the viral proteome, including spike proteins, with the recognition of the spike RBD [33,34,35,36,37,38,39,40,41].

## 2. Results

### 2.1. Mapping of IgG Epitopes within SARS-CoV-2 Spike Protein

The coding sequence of the spike protein of SARS-CoV-2 was represented by a library of 15 mer peptides offset by five amino acids synthesized directly onto a cellulose membrane. Immunodominant epitopes were identified by an SPOT synthesis analysis using a pool of hospitalized individuals with severe COVID-19 infections and chemiluminescent imaging of bound IgG (Figure 1). Signal intensities were normalized to the maximum signal, and an intensity level cut-off of 30% was used to define the epitopes. Forty-one linear B-cell epitopes ranging from five to fifteen amino acids were identified (Table 1). Nineteen epitopes were located in the S1 domain, seven in the N-terminal domain (NTD), seven in the receptor binding domain (RBD), two in SD1, and another three in SD2. One epitope, TQTNSPRRAR, was detected in the core region of the furin cleavage site (685RS686). The S2 fragment housed twenty-two (22) epitopes that included an epitope encompassing the TMPRSS2 cleavage site, LPDPSKPSKRSFIED (815RS816), and fusion peptide 1 (816–837).

Three epitopes were found in the first Heptad repeat (HR1; 920–970) and two in Heptad repeat 2 (HR2; 1163–1202). No epitopes were localized to the transmembrane domain (TMD), and there was only one in the C-terminal after the TMD. Among the epitopes with signals greater than 80%, two (2) were located in the RBD (GKIADYNYKL and PLQSYGFQPTGVGY) and one (1) in the S2 domain (LPPLL). Following their localization, each epitope sequence was searched using the Basic Local Alignment Search Tool for proteins (BLASTp) and multiple sequence alignments restricted to the coronavirus family. Using four consecutive and identical amino acids as the minimum binding site for an epitope, most epitopes showed no commonality to endemic coronaviruses, suggesting that some epitopes are specific to SARS-CoV-2 (Appendix A).

### 2.2. Cross-Reactivity with Anti-DENV Antibodies

Previous evidence has suggested that the SARS-CoV-2 S1-RBD could be recognized by antibodies in patients infected with DENV. When we expanded the BLASTp parameters to include the Dengue virus (Taxid: 12,637), twenty-one sequences presented a cross-reactivity potential based on the above criteria (Appendix A). Next, the reactivity of serum samples (*n* = 45) from patients with antibodies from DENV infections before the COVID-19 pandemic were evaluated in a commercial ELISA utilizing the spike protein and nucleoprotein of SARS-CoV-2 as targets for capturing antibodies. Figure 2a presents the reactivity index of the individual samples and shows that five samples from DENV patients were positive (11%). Pre-pandemic serum samples were also tested in a commercial ELISA for DENV (1–4). DENV antibodies (IgG) were highly seropositive among healthy blood donors (20; 71.4%), and, as expected, the highest seropositive sera were found in the small library of DENV-positive pre-pandemic sera (37; 92.5%) (Appendix A).

A SPOT synthesis analysis was used to determine if the reactivity of the pre-pandemic DENV serum could bind to the SARS-CoV-2-specific epitopes in its spike protein (Figure 2c). From the forty-one previously identified epitopes, ten presented a normalized signal intensity >50% to pre-pandemic DENV serum, defined to be cross-reactive based on a statistical analysis. Three cross-reactive regions were located in S1-NTD, two in the RBD region, one in S1/SD1, one in the furin cleavage site, and the remaining three in the S2 domain.

### 2.3. Bioinformatic Analysis

An open question was whether the sequence similarity between the SARS-CoV-2 spike protein and DENV proteins alone could explain the observed cross-reactivity. Additional BLASTp searches were performed with the ten epitope sequences displaying cross-reactivity to identify DENV-1–4 proteins or polyproteins (Table 2). Searches were conducted for short input sequences of at least four consecutive amino acids to identify mimetic peptides in the DENV. Most spike epitopes (*n* = eight) had a sequence of at least four amino acids identical to a Dengue protein. The other two epitopes presented sequence gaps with a lower level of sequence identity. The identified sequences have a high level of conservation among the VoCs (Appendix A) and a low level of identification against the endemic coronaviruses (Appendix A).

Sequence identity per se does not signify cross-reactivity, as antibody interaction also depends on structure conformation and accessibility to antigens. Therefore, potential cross-reactive sequences were located within the spike protein monomer and trimer, and the solvent accessibility area (SASA) was calculated (Figure 3a), which is correlated to the spatial arrangement and exposure of residues to the solvent. In a side view of the spike trimer (Figure 3b), several surface-exposed residues in cross-reactive epitopes were localized, which supported their potential to interact with antibodies (Figure 3b). Likewise, highly surface-exposed areas were present at the top of the trimeric spike protein (Figure 3c). The FPQS residues are in a buried surface area of the spike protein with low solvent accessibility and are unlikely to represent a cross-reactive site. Within the Ns5 and envelope protein E of DENV2 and the NS3 helicase of DENV4, searches of the epitope sequences in the protein databank revealed similar epitope sequences with surface exposure (Figure 3d).

### 2.4. Pre-Pandemic DENV Sera Display Antibody-Dependent Enhancement In Vitro

After identifying cross-reactive sites, the question remained as to whether the presence of antibodies formed during a DENV infection could potentiate antibody-dependent enhancement of SARS-CoV-2 infections. To test this hypothesis, monocytes that express Fc receptors (FcRs) were used as a model system for infection with SARS-CoV-2 in the presence of a pool of pre-pandemic DENV-positive sera over a two-fold serial dilution series. Incubation of the virus with pre-pandemic DENV-positive sera with a dilution ratio of one to four induced a significant increase (*p* = 0.014) in the viral load of monocytes that was 1.7 times the control loads (Figure 4a). Measurements of lactate dehydrogenase for cell death showed no significant loss of monocytes from the increase in viral entry, which was expected since SARS-CoV-2 replication is aborted in monocytes (Figure 4b). The neutralization capacity of a DENV patient serum pool was also evaluated by a plaque assay using Calu-3 cells. No neutralization (Figure 4c) or significant cell death (Figure 4d) was measured for the pre-pandemic Dengue-positive pool compared to the COVID-19 control and healthy human sera. 

### 2.5. Antibody Binding to Peptides from Spike Protein

To evaluate the cross-reactivity sequences in spike proteins, we synthesized three peptides with a strong signal in the SPOT synthesis and solvent accessibility analyses (LMDLEGKQG NFKNLR, LGVYYHKNNK, and GKIADYNYKL). The peptides were used in a competitive ELISA, and different concentrations were pre-incubated with a pool of DENV-positive samples (Figure 5). The samples were used in the NovaLisa Dengue IgG kit. The only peptide that significantly reduced antibody binding was LMDLEGKQGNFKNLR, which contained the MDLE sequence from the DENV envelope protein. Compared to the untreated control, the peptide concentrations of 500 and 250 ng/well were reduced by approximately 23% and 20%, respectively. Cross-reactivity with other coronaviruses is plausible. However, multiple sequence alignments of spike proteins with other coronaviruses and endemic viruses were performed. Similarities were found for Bat-Cov and Sars-CoV, but not for other coronaviruses. A peptide ELISA using the LMDLEGKQGNFKN LR sequence showed no significant difference between the pre-pandemic, DENV-positive pre-pandemic, and COVID-19-vaccinated samples (Appendix A). The results highlight that this sequence elicits non-specific antibody reactions.

## 3. Discussion

The World Health Organization recently declared the end of the COVID-19 public health emergency. Yet, the emergence of variants that display increased transmissibility, are refractory to previously neutralizing antibodies, or both shows the importance of continued studies on the humoral response to SARS-CoV-2 and other pathogens to develop next-generation vaccines and therapies. Here, we began by identifying the immunodominant epitopes in the spike protein of SARS-CoV-2 that are recognized by antibodies in the serum of hospitalized COVID-19 patients. Employing an SPOT synthesis peptide microarray, a total of 41 epitopes were identified. These epitopes were distributed across different domains of the spike protein, including S1, NTD, RBD, SD1, SD2, furin cleavage site, S2 fragments, and heptad repeats 1 and 2. Some of these epitopes show partial or total sequence similarities to some epitopes reported in previous studies on SARS-CoV-2 [42,43,44,45,46,47], indicating consistency between the techniques and serum panels used, which is important for defining immunodominant epitopes across multiple populations stimulated by natural infections and vaccinations. Screening immunodominant epitopes is important to understanding the immune response, supporting vaccine development by identifying key antigenic targets that elicit strong and specific immune responses, making accurate diagnoses, and understanding the immune escape. The epitopes of other viral proteins are equally crucial; while mutations in spike proteins are linked to immune evasion and infectivity in Omicron variants, their pathogenicity is also correlated with NSP6 [48].

Epitope mapping is also a strategic method for developing broadly neutralizing antibodies or nanobodies for prophylactic, diagnostic, and therapeutic use [49]. Neutralizing epitopes are predominantly located within the NTD and RBD, and antibodies binding to the RBD can account for over 90% of neutralizing activity in convalescent sera [50]. Neutralizing antibodies (nAbs) can also target the S2 stem helix (SH) and S2 fusion peptide (FP) regions [51,52,53]. The immune evasion of variants of concern is driven by mutations in the spike protein that threaten natural and vaccine-induced immunity. However, conserved pan-variant epitopes that confer broad neutralization have great potential for therapy and vaccine design [26,54,55,56,57,58].

The binding mode of nAbs can be divided into four main classes depending on the location of their epitopes in the spike protein [50]. The RBD region is targeted by class 1 and 2 antibodies which include the RBM and can compete with ACE2 binding. The major IgG epitopes CV19/SG/13huG and CV19/SG/14huG are located within this region. Class 3 RBD antibodies bind to regions flanking the ACE2-binding region, and the epitopes CV19/09huG and CV19/SG/12huG could be involved in the binding of nAbs. Importantly, this region contains highly conserved residues in the SARS-CoV and SARS-CoV-2 RBDs which could confer broad cross-reactivity as observed with antibody S30924, which also neutralized the SARS-CoV-2 VoC Omicron B.1.1.529 [59]. The epitope recognized by class 4 antibodies is highly conserved in the RBD but does not directly block ACE2–RBD binding [50]. This cryptic region spans from residues 369 to 385, but our study found low reactivity for any epitope in this region.

Changes to epitope sequences in the NTD and RBD will likely contribute to immune escape by variants. In contrast, the neutralizing epitopes in the S2 subunit are more conserved across variants [60]. However, low reactivity was observed for the S2 SH region, which spans 14 residues (1146–1159) and is conserved across beta-CoVs [61]. The S2 FPs are also highly conserved among all coronaviruses, suggesting an antibody targeting this region could display broad-spectrum activity. The epitope CV19/SG/26huG (LPDPSKP SKRSFIED) was identified in the S2 FP1, which overlaps with the ‘RSFIEDLLF’ motif bound by several human monoclonal antibodies isolated from convalescent patients that bind the [62,63]. The R815 conserved residue is the S2′ site of TMPRSS2, and by targeting this region, antibodies would interfere with cleavage and inhibit the membrane fusion of S protein [63].

SPOT synthesis analyses are also useful for identifying cross-reactive epitopes, which are a significant concern in developing serological tests, and SARS-CoV-2 is no exception. Although SARS-CoV-2 proteins exhibit low sequence similarity to other human coronaviruses (HCoVs), previous evidence has suggested that the response to SARS-CoV-2 may be influenced by earlier exposures to other coronaviruses and other endemic viruses, including respiratory viruses [64]. Fifteen highly antigenic epitopes against pathogens, self-proteins, and common human viruses in a cohort of pre-pandemic individuals naïve to SARS-CoV-2 showed cross-reactivity with the spike protein of SARS-CoV-2 [65]. This points to a potential limitation of linear B-cell epitope mapping with the presence of cross-reactive antibodies in serum samples obtained from individuals with unknown clinical histories. Only eight epitopes identified in this study were non-specific for SARS-CoV-2 [65].

By confirming the results with bioinformatics, it is possible to identify cross-reactive epitope sequences based on a minimum antibody binding site of four consecutive amino acids. Here, ten sequences exhibited potential cross-reactivity with the Dengue virus. Dengue fever is a vector-borne viral disease caused by the flavivirus Dengue virus which has four serotypes, each with three structural and seven non-structural proteins [66,67]. It is endemic to tropical regions worldwide which are home to approximately 2.5 billion people, and there are estimated to be 400 million yearly cases globally. Previous data have shown that pre-pandemic Dengue patient sera contain cross-reactive antibodies for SARS-CoV-2 that can cause false-positive SARS-CoV-2 serology results and could partially explain the low specificity of some serological tests [68]. A recent study highlighted the antigenic similarity between the SARS-CoV-2 S1-RBD regions and DENV proteins (E and NS1), which would allow us to induce cross-reactive antibodies after SARS-CoV-2 S1-RBD immunization in vivo using an experimental animal model. 

The cross-reactivity between DENV and SARS-CoV-2 occurs in diagnostic tests, and some studies have shown a minimal risk of serological cross-reactivity between DENV and SARS-CoV-2 IgG antibodies when employing serological assays [69]. However, other works have demonstrated a higher profile of cross-reactivity of SARS-CoV-2 antibodies, but not neutralizing antibodies, among individuals previously exposed to DENV. Interestingly, assays utilizing spike S1 as the antigen detected more positives among primary DENV infections than in-house ELISAs using receptor binding domain (RBD) proteins [70]. It was demonstrated that DENV cross-reactive antibodies might cause false-positive results in Dengue serological tests [71] and potentially hinder Dengue infections [8].

Using commercial ELISA kits for COVID-19 and DENV, our results demonstrated that cross-reactivity exists between antibodies generated in response to SARS-CoV-2 and DENV. These findings highlight the weakness of full-length viral proteins in serological tests containing a mixture of definable specific and non-specific linear B-cell epitopes. 

This issue is particularly relevant in SARS-CoV-2 and DENV due to the potential for cross-reactive DENV antibodies to facilitate antibody-dependent enhancement (ADE). ADE has been observed in various viral infections, including DENV, West Nile virus, Ebola virus, measles, respiratory syncytial virus, and human immunodeficiency virus (HIV) [12,15,20]. Also, in SARS-CoV ADE, vaccine preclinical trials in animal models have been reported [72]. In the context of SARS-CoV-2, ADE has been debated since the pandemic’s beginning, particularly concerning the potential risk associated with vaccination, convalescent serum therapy, and the emergence of new variants.

ADE has been extensively studied in Dengue fever. Individuals with pre-existing immunity against one serotype of DENV are at a higher risk of developing severe Dengue shock syndrome (DSS) or Dengue hemorrhagic fever (DHF) when infected with a different serotype [73]. This fact is attributed to antibodies with low neutralizing activity but that can facilitate viral entry into macrophages via Fc receptors [74]. ADE-driven infection of macrophages can increase viral replication and load [75]. This clinical concern has challenged the development of a safe and effective Dengue vaccine that induces a balanced immune response against all serotypes to avoid ADE [76].

Fortunately, unlike DENV, SARS-CoV-2 does not replicate productively in macrophages. Macrophages can use phagocytosis to counter the virus, but this does not result in productive infection. However, SARS-CoV-2’s entry into immune cells leads to an abortive infection followed by host cell pyroptosis, the release of proinflammatory cytokines, and the activation of inflammasomes [77,78].

Apart from FcR-dependent ADE, there is also evidence of FcR-independent ADE in SARS-CoV-2 infections. Antibodies that recognize specific binding domains on the SARS-CoV-2 spike protein can induce structural changes that facilitate viral entry and infection. Some antibodies can mediate the ADE of diseases in vitro in an FcR-independent manner [79]. Human mAbs against SARS-CoV-2 spike protein were found to enhance the virus infection in vitro by the FcγR-mediated pathway [17]. In addition, ADE was reported to neutralize mAbs using SARS-CoV-2 pseudovirus infections on FcγRIIB-expressing B-cells; the bivalent interaction of antibodies with S-trimer RBDs enhanced the activity of the SARS-CoV-2 pseudovirus on FcγRIIB-expressing B-cells [80]. A study revealed that FcγR- and/or C1q-mediated ADE was detected in 50% of IgG-positive sera.

Interestingly, most of these sera also exhibited neutralizing activity without FcγR and C1q. ADE antibodies were found in 41.4% of acute COVID-19 patients, suggesting the potential for ADE to promote virus replication even in the early phase of infection [18]. The authors proposed that C1q-mediated ADE may occur in the respiratory tissues of COVID-19 patients, and possibly ADE plays a role in the exacerbation of the disease [18].

Different factors contribute to antibody efficacy, depending on titers, affinity viral epitopes, and stoichiometry. The production of neutralizing antibodies is considered the primary goal of vaccination. Still, cross-reactive or non-neutralizing antibodies may also be produced, which can impact the severity of the infection. The presence of cross-reactive antibodies or non-neutralizing antibodies may increase disease severity through ADE [81]. Although ADE has not been observed on a widespread scale in COVID-19 cases, the risks associated with ADE include the potential for more severe disease outcomes, increased viral replication, and exacerbated immune responses.

In vitro ADE does not always correlate with enhanced infection in vivo because other antibody functions could play a role in viral clearance, such as antibody-dependent cell-mediated cytotoxicity (ADCC) and complement-dependent cytotoxicity (CDC). Previous studies with SARS-CoV-2 antibodies have shown an in vitro enhancement of infections but in vivo protection in animal models [72]. Although a small effect of ADE was evidenced on pre-pandemic DENV-positive serum in vitro, there is no proof that this event could have happened in vivo, enhancing the infection. 

Evidence of a rare effect of ADE has been reported in preclinical animal models infused with antibodies [82]. Additionally, a mouse monoclonal antibody (mAb RS2) against spike protein increased the viral load in the respiratory tracts of animals in post-exposure prevention studies [83]. Interestingly, a competitive ELISA showed that antibodies competing with the mAb RS2 epitope are significantly higher in the plasma of patients with severe COVID-19 infections than those in patients with mild infections or vaccinated individuals [83]. This finding underscores the potential urgency in understanding the role of antibodies with a similar binding of mAb RS2 in developing more severe COVID-19 infections. Recently, a study analyzed ADE in patients exposed to Middle East respiratory syndrome coronavirus (MERS-CoV); 56% demonstrated ADE against an SARS-CoV-2 pseudo-virus. However, subsequent exposure to SARS-CoV-2 vaccination diminished this ADE activity [84]. However, despite the concern raised in vaccine development, it is important to note that ADE has not been significantly demonstrated in COVID-19 vaccines [85]. This finding provides reassurance and alleviates any concerns about ADE in the context of COVID-19 vaccination.

Highly fucosylated antibodies of vaccines do not cause ADE, but fucosylated antibodies produced in acute primary infections or convalescent sera can induce it [77]. It is essential for vaccine development to balance ADE and neutralizing antibodies. Candidates for vaccines must elicit an immune response that produces neutralizing antibodies capable of successfully preventing viral entry and the specific recognition of pathogen antigens. Epitope selection and studies of cross-reactive antibodies are essential for the rational design of vaccines, antibody therapy, and serological diagnostic tests. ELISA tests have demonstrated cross-reactivity sequences in spike proteins, which could elicit the non-specific binding of antibodies that are not restricted to DENV infections. These findings may contribute to the development of recombinant proteins that lack these cross-reactive sites to increase the specificity of diagnostic tests and epitope-based vaccines.

Similarly, other studies have highlighted that pre-existing cross-reactive IgA [45] and IgG serum antibodies against spike proteins were detectable in pre-pandemic cohorts [86]. Our results highlight that IgG spike cross-reactive sequences could be recognized non-specifically by antibodies drawn from pre-pandemic sera. This result opens new avenues for studying the influence of cross-reactive sites on immunodiagnostics, disease, and vaccine development.

## 4. Materials and Methods

### 4.1. Patient Samples

Serum panels were collected by venipuncture using vacuum tubes containing gel separators. After being collected, the tubes were gently inverted to mix the blood with the gel. Subsequently, the samples were centrifuged at 3.000 rpm for 10 min, and serum was separated using a pipette. Serum samples (30) were obtained from individuals who received a single dose of the Oxford/AstraZeneca vaccine, and 7 serum samples were collected from individuals who received four heterologous doses, all primed with Oxford/AstraZeneca and booster of Pfizer Spike mRNA vaccine or Janssen (Ad26.COV2-S). Also, Dengue (1–4) pre-pandemic serum (*n* = 45) samples were utilized. Additional negative controls (*n* = 28) included sera collected before the pandemic from healthy individuals from blood bank donors (HEMORIO, Rio de Janeiro, Brazil). Patient privacy was maintained by excluding identifying information. Serum samples from individuals with Dengue were collected before the onset of the COVID-19 pandemic and generously provided by the Laboratory of Flavivirus of the Oswaldo Cruz Institute (FIOCRUZ, Rio de Janeiro, Brazil). 

### 4.2. B-Linear Epitope Mapping

The complete sequences of the spike protein of SARS-CoV-2 were retrieved from the UniProt database (http://www.uniprot.org/: accessed on 27 January 2020). Microarrays of peptides and a pool of human COVID-19 patient sera (*n* = 12) were used to map linear B-cell IgG epitopes using Auto-Spot Robot ASP-222 synthesizer (Intavis Bioanalytical Instruments AG, Köln, Germany) according to a previous SPOT synthesis protocol [87].

### 4.3. Peptide Synthesis

SARS-CoV-2 10 mer and 15 mer peptides (LMDLEGKQGNFKNLR, LGVYYHKNN K, and GKIADYNYKL) were chosen for synthesis using the F-moc strategy in a synthesizer machine (MultiPep-1 CEM Corp, Charlotte, NC, USA), using methods previously described [86].

### 4.4. Enzyme-Linked Immunosorbent Assay (ELISA)

Using the manufacturer’s instructions, ELISA assays were performed using a confirmatory commercial COVID-19 IgG kit (Dia. Pro, Giovanni Milan, Italy). Each microplate strip has its wells coated with nucleocapsid and spike-specific antigens, making it possible to map the serological response to the different IgG types produced. To detect DENV (1–4) IgG antibodies, a SERION ELISA classic/antigen Dengue Virus IgG commercial kit (#ESR114G; Virion/Serion GmbH, Würzburg, Germany) was used. Tests were performed following the manufacturer’s instructions. Results are represented as reactivity index calculated using the formula (IR = sample absorbance/cut-off). 

Competitive ELISA was performed using NovaLisa ELISA IgG kit (DENG0120) to detect DENV antibodies. A pooled serum sample from eight DENV-positive patients was prepared and diluted to a 1:100 concentration utilizing the kit’s diluent. The serum was preincubated for 1 h at room temperature with each of the three SARS-CoV-2 spike peptides (LMDLEGKQGNFKNLR, LGVYYHKNNK, and GKIA DYNYKL) at six different concentrations: 500, 250, 125, 62.5, 31.2, 15.6, and 7.8 ng/well. Serum without any peptide added served as the control. Following preincubation, the ELISA was performed according to the manufacturer’s instructions. The values were recorded and the control values (serum without peptides) were compared to values from serum preincubated with different peptides and concentrations. This comparison allowed for the evaluation of the binding efficiency and inhibition potential of the peptides.

In-house peptide ELISA was performed as described previously [88]. Briefly, 500 ng of peptides in a coating buffer (50 mM carbonate-bicarbonate buffer, pH 9.6) was added to Immunolon 4HB plates (Immunochemistry Technologies, Bloomington, MN, USA) overnight at 4 °C. After washing with PBS-T (phosphate-buffered saline plus 0.05% Tween^®^ 20), plates were incubated for 1 h at 37 °C with 1% bovine serum albumin (BSA) (200 µL) in PBS-T to block free binding sites. Subsequently, following the dilution, sera were diluted (1:25) in coating buffer and PBS-T with 1% BSA (1:25). Then, 100 µL of the diluted sera was applied to plates, which were incubated for 1 h at 37 °C. Following washing with PBS-T, plates were incubated with 100 µL of anti-human IgG HRP (1:10.000) for 1 h. Binding antibodies were revealed by adding 3,3′,5,5′-tetramethyl benzidine (1-Step™ Ultra TMB-ELISA, Science Biotech Ltd. Lages, Brazil), which was added for 15 min, and the reaction was stopped after a few minutes with 0.5 M sulfuric acid. Optic density was evaluated at 450 nm within 2 h of adding the benzindine. 

### 4.5. *In Silico* Analysis

BLASTp (http://blast.ncbi.nlm.nih.gov) was used to find cross-reactive sequences in SARS-CoV-2 epitopes that match the Dengue virus (Taxid: 12,637). The website was accessed on 16 May 2023. The BLASTp algorithm parameters were set as follows: expect threshold of 30,000, word size of 2, and PAM30 matrix. Moreoever, gap cost was set to existence = nine and extension = 1, the compositional parameter was set to no adjustment, and the low-complexity filter was disabled and adjusted for short input sequences. The results from BLASTp analysis were screened and filtered for at least four contiguous and identical amino acid residues with the DENV (1–4) peptides with no gap and no mismatched residues. Additionally, peptides were screened using Protein Information Resource (PIR; https://research.bioinformatics.udel.edu/peptidematch/index.jsp, accessed on 10 September 2023). Spike trimer protein in a closed state was retrieved from I-Tasser (https://zhanggroup.org/I-TASSER/, accessed on 5 November 2022). Annotation of epitopes and cross-reactive sites was performed using Chimera X [89]. The accessible surface area was performed by using the accessibility calculation for protein (ver. 1.2) online server that calculates the solvent-accessible surface areas of spike protein (P0DTC2) amino acids (http://cib.cf.ocha.ac.jp/bitool/ASA/, accessed on 10 October 2023).

### 4.6. Cells, Viruses, and Reagents

African green monkey kidney cells (Vero, E6 cell; ThermoFisher, Waltham, MA, USA) and human lung epithelial cell lines (Calu-3) were expanded in high-glucose DMEM with 10% fetal bovine serum (FBS; Sigma-Aldrich, St Louis, MO, USA), with 100 U/mL penicillin and 100 μg/mL streptomycin (Pen/Strep; ThermoFisher, Waltham, MA, USA) at 37 °C in a humidified atmosphere with 5% CO_2_. Peripheral blood mononuclear cells (PBMCs) were isolated by density gradient centrifugation (Ficoll-Paque, GE HealthCare, Chicago, IL, USA) from buffy coat blood preparations from healthy donors. PBMCs (2 × 10^6^ cells) were plated into 48-well plates (NalgeNunc Int Corrp, Rochester, NY, USA) in RPMI-1640 with 5% inactivated male human AB serum (Sigma-Aldrich, St Louis, MO, USA) for 3 h. Non-adherent cells were removed, and monocytes were maintained in (low-glucose) DMEM with 5% human serum, 100 U/mL penicillin, and 100 μg/mL streptomycin. The purity of monocytes was above 90%, as determined by flow cytometry (FACScan; Becton Dickinson, Juiz de Fora, Brazil) using anti-CD3 (BD Biosciences, Mississauga, ON, Canada) and anti-CD14 (BD Biosciences, Mississauga, ON, Canada) antibodies. SARS-CoV-2 (GenBank accession no. MT710714) was expanded in Vero E6 cells. Viral isolation was performed after a single passage in cell culture in 150 cm^2^ flasks with high-glucose DMEM plus 2% FBS. Observations for cytopathic effects were performed daily and peaked 4 to 5 days after infection. All procedures related to virus culture were handled in biosafety level 3 (BSL3) multiuser facilities, according to WHO guidelines. Virus titers were determined as plaque-forming units (PFU/mL), and virus stocks were kept in −80 °C ultra-low-temperature freezers. 

### 4.7. Infections and Virus Titration

Infections were initiated with SARS-CoV-2 at MOI of 0.1 in low (monocytes)- or high (Calu-3)-glucose DMEM without serum. Viral input was incubated with diluted serum samples for 15 min before exposure to cell culture. After 1 h, the unbound virus was removed, and cells were washed and incubated with a complete medium. For virus titration, monolayers of Vero E6 cells (2 × 10^4^ cells/well) in 96-well plates were infected with serial dilutions of supernatants containing SARS-CoV-2 for 1 h at 37 °C. Semi-solid high-glucose DMEM medium containing 2% FBS and 2.4% carboxymethylcellulose was added, and cultures were incubated for three days at 37 °C. Then, the cells were fixed with 10% formalin for 2 h at room temperature. The cell monolayer was stained with 0.04% solution of crystal violet in 20% ethanol for 1 h. Plaque numbers were scored with at least three replicates per dilution by independent readers blinded to the experimental group, and the virus titers were determined by plaque-forming units (PFU) per milliliter. Experimental procedures involving human cells from healthy donors were performed with samples obtained after their written informed consent was given.

### 4.8. Molecular Detection of Viral RNA Levels

According to the manufacturer’s instructions, total RNA was extracted from cells using QIAamp Viral RNA kit (Qiagen Sciences Inc., Germantown, MD, USA). Quantitative RT-PCR was performed using QuantiTect Probe RT-PCR Kit (Qiagen Sciences Inc., Germantown, MD, USA) in a StepOnePlus™ Real-Time PCR System (ThermoFisher, Waltham, MA, USA). Amplifications were performed in 15 µL reaction mixtures containing 2X reaction mix buffer, 50 µM of each primer, 10 µM of the probe, and 5 µL of RNA template. Primers, probes, and cycling conditions followed the protocol the Centers for Disease Control and Prevention (CDC, Atlanta, GA, USA) recommended to detect SARS-CoV-2. A standard curve method was employed for virus quantification. The housekeeping gene RNAse P was amplified to reference the cell quantities assayed. The Ct values for this target were compared to calibrations obtained from 10^2^ to 10^7^ cells.

### 4.9. LDH Measurement

Cell death was determined by proxy using the liberated lactate dehydrogenase (LDH) activity level in supernatants using CytoTox^®^ Kit (Promega, Madison, WI, USA). Supernatants were centrifuged at 5000 rpm for 1 min before an assay to remove cellular debris.

## 5. Conclusions

We utilized an SPOT synthesis analysis to map IgG immunodominant epitopes in the spike protein of SARS-CoV-2. The distribution of these epitopes across different domains of the spike protein highlights the complexity of the humoral immune response and the potential for diverse antibody interactions. Further, our findings suggest that individuals with a previous DENV infection may harbor cross-reactive antibodies that can lead to ADE in vitro. The observations that pre-formed antibodies in pre-pandemic serum against peptides from spike proteins have many implications in immunodiagnostics. Also, the sequence similarities to previously reported epitopes in SARS-CoV-2 highlight the importance of understanding the immunodominance of epitopes in the population. This knowledge can aid in the development of broadly neutralizing antibodies or nanobodies for prophylactic, diagnostic, and therapeutic purposes. The presence of conserved pan-variant epitopes provides opportunities to design effective vaccines and therapeutics that can target multiple SARS-CoV-2 variants. In conclusion, our study sheds light on the cross-reactivity of IgG epitopes and the potential for antibody-mediated enhancement in vitro in SARS-CoV-2 and previous viral infections. This knowledge has important implications for understanding the immune response to SARS-CoV-2 and developing strategies to combat the ongoing COVID-19 pandemic.

## 6. Patents

Patent applications were filed on 5 June 2020 for the epitope targeting of SARS-CoV-2 and the construction of chimeric proteins (provisional patent applications BR1120210214011 (Brazil), US17638108 (USA), EP4039696 (Europa), CN114258399 (China), and IN202217 004847 (India) to D.W.P-Jr., A.M.D., P.N.-P., and S.G.D.-S., Oswaldo Cruz Foundation). 

## Figures and Tables

**Figure 1 ijms-25-08180-f001:**
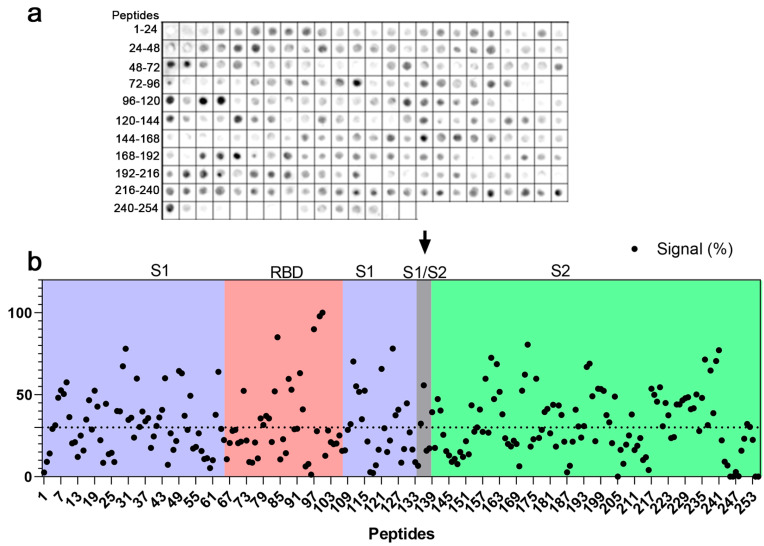
Map of linear IgG B-cell epitopes in SARS-CoV-2 spike protein. An SPOT synthesis analysis using a library of 15 mer peptides with an overlap of 5 residues to represent the S protein-coding sequences synthesized directly onto a cellulose membrane. (**a**) Chemiluminescent image of peptides recognized by antibodies after probing in a serum pool from hospitalized individuals with severe COVID-19 infections. Each spot represents a peptide. (**b**) Graphical representation of the signals measured at each peptide spot and normalized to the maximum signal. Peptide sequences are displayed with an intensity level above 30% (dashed line), defined as a positive reaction. Peptides were synthesized following the primary sequence of spike protein and highlighted in receptor binding domain (RBD), as well as S1, S1/S2 (furin cleavage site; arrow), and S2 domains.

**Figure 2 ijms-25-08180-f002:**
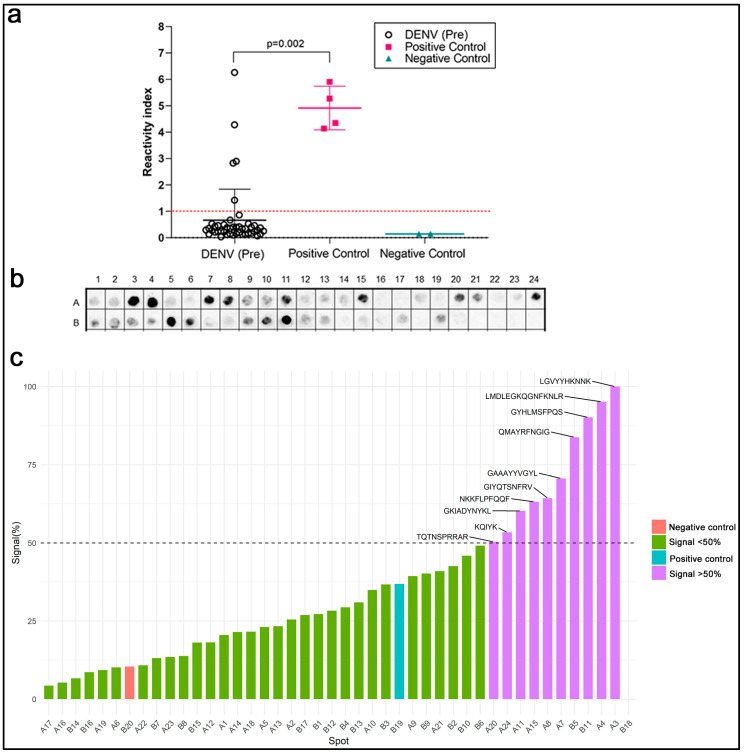
Cross-reactivity between DENV and SARS-CoV2- infections. (**a**) Commercial SARS-CoV-2 ELISAs using sera from pre-pandemic DENV-positive patients and kit negative and positive controls. (**b**) Chemiluminescent image of an SPOT synthesis analysis of the 41 epitopes identified in the SARS-CoV-2 spike protein using a serum pool from pre-pandemic DENV-infected patients (*n* = 8), revealing reactive IgG antibodies. Each spot represents one of the 41 identified epitopes displayed in rows (A and B) and columns (1–24). (**c**) Graphic representation of quantifying signal intensities normalized to the maximum signal. An intensity level above 50% was defined as reactive. A Kruskal–Walli’s test was applied to identify statistical differences, followed by Dunn’s multiple comparison tests. A *p* < 0.05 was considered to indicate a significant difference.

**Figure 3 ijms-25-08180-f003:**
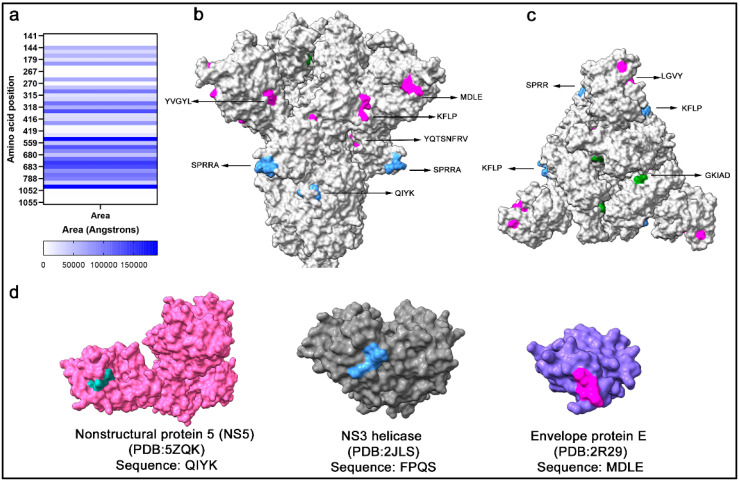
Solvent accessible area of cross-reactive SARS-CoV-2 spike peptides. (**a**) Data represent the exposed area for the calculation of SASA for each residue. (**b**) Cross-reactive residues are marked in spike structures in the S1 domain (magenta), RBD (green), and S2 domain (blue). (**c**) The top view of the spike trimer shows clusters of cross-reactive residue sites in the RBD, S1 domain, and S2 domain. (**d**) Structural model of DENV-2 NS5 (pink), NS3 helicase (gray), and envelope protein E (blue) with the identical residues in SARS-CoV-2 spike QIYK (cyan), FPQS (blue), and MDLE (magenta), respectively.

**Figure 4 ijms-25-08180-f004:**
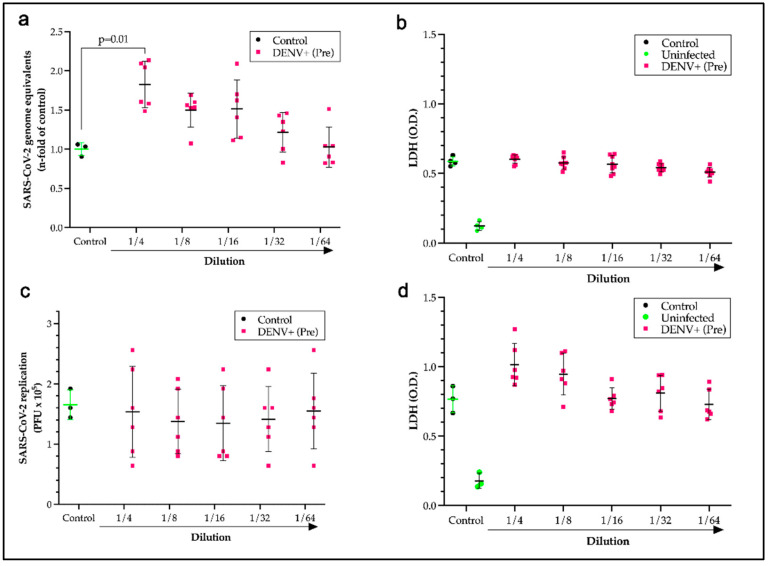
Antibody-dependent enhancement of SARS-CoV-2 infections of monocytes without cell death or neutralization by pre-pandemic DENV-positive sera. A monocyte in vitro infection assay with SARS-CoV-2 virus preincubated with vehicle (control), showing a 2-fold serial dilution series of a pool of pre-pandemic DENV patient sera or normal sera (uninfected) measuring viral load by qPCR (**a**) and lactate dehydrogenase (LDH) for cell death (**b**). An in vitro infection assay with Calu-3 cells with SARS-CoV-2 virus preincubated with vehicle (control), showing a 2-fold serial dilution series of a pool of pre-pandemic DENV patient sera or normal sera (uninfected) measuring: (**c**) SARS-CoV-2 replication levels as plaque-forming units (PFUs) or LDH levels as a cell death indicator (**d**). Data show the individual data from six experiments and the mean and standard deviation. Kruskal–Walli’s test was applied to identify statistical differences, followed by Dunn’s multiple comparison tests. Significant differences were considered with *p* < 0.05.

**Figure 5 ijms-25-08180-f005:**
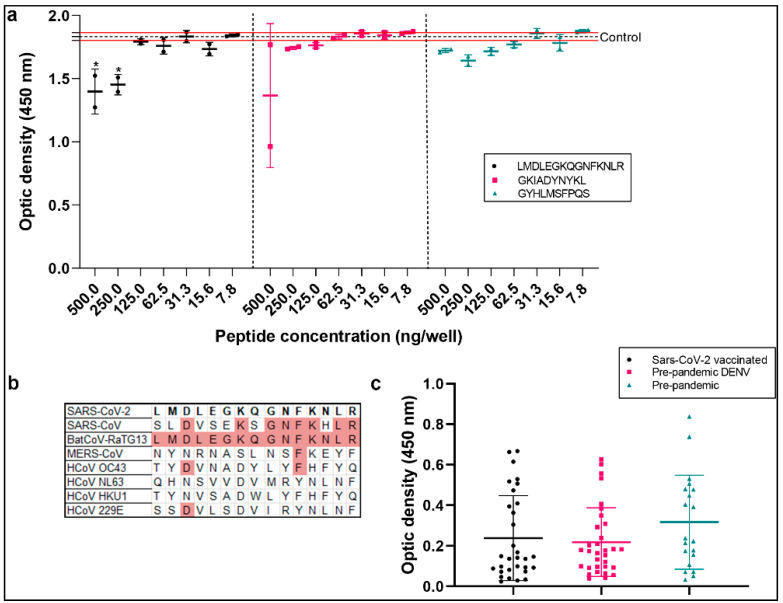
Analysis of spike cross-reactive peptides. (**a**) Competitive ELISA using DENV IgG test performed using a pool of DENV-positive samples pre-incubated with peptides from spike protein. The control group was tested in triplicate at 1:100 without peptides, and the data show the mean (dotted line) and standard deviation (red lines). (**b**) Multiple sequence alignment of spike protein (P0DTC2) and other coronaviruses (SARS-CoV-1 (P59594), MERS-CoV (K9N5Q8), HCoV-OC43 (P36334), HCoV-NL63 (Q6Q1S2), HCoV-HKU1 (Q14EB0), HCoV-229E (P15423), and BatCoV-RaTG13 (A0A6B9 WHD3)); similar amino acids are highlighted in red. (**c**) In-house peptide ELISA performed with peptide LMDLEGKQGNFKFKNLR and a cohort of pre-pandemic sera (*n* = 17), individuals vaccinated with the first dose of Oxford–AstraZeneca and heterologous doses (*n* = 31), and DENV-positive pre-pandemic group (*n* = 32). A one-way ANOVA was applied to identify statistical differences, followed by Dunnet’s multiple comparison tests. *p* < 0.05 was considered a significant difference (* *p* < 0.05).

**Table 1 ijms-25-08180-t001:** IgG epitopes mapped in SARS-CoV-2 spike protein using COVID-19 patient sera.

Code	aa Position	Sequence	Domain
CV19/SG/01huG	86–100	FNDGVYFASTEKSNI	S1/NTD
CV19/SG/02huG	111–125	DSKTQSLLIVNNATN	S1/NTD
CV19/SG/03hu1G	141–150	LGVYYHKNNK	S1/NTD
CV19/SG/04huG	176–190	LMDLEGKQGNFKNLR	S1/NTD
CV19/SG/05huG	211–220	NLVRDLPQGF	S1/NTD
CV19/SG/06huG	246–256	RSYLTPGDSSS	S1/NTD
CV19/SG/07huG	261–270	GARVEY	S1/NTD
CV19/SG/08huG	311–320	GIYQTSNFRV	S1/RBD
CV19/SG/09huG	355–364	KRISNCVADYSVLYN	S1/RBD
CV19/SG/10huG	396–404	YADSFVIRGD	S1/RBD
CV19/SG/11huG	416–425	GKIADYNYKL	S1/RBD
CV19/SG/12huG	441–450	LDSKVGGNYN	S1/RBD-RBM
CV19/SG/13huG	461–470	LKPFERDIST	S1/RBD-RBM
CV19/SG/14huG	491–505	PLQSYGFQPT	S1/RBD-RBM
CV19/SG/15huG	556–564	NKKFLPFQQF	S1/SD1
CV19/SG/16huG	571–575	DTTDAVRDPQ	S1/SD1
CV19/SG/17huG	606–615	NQVAVLYQDV	S1/SD2
CV19/SG/18huG	626–635	ADQLTPTWRV	S1/SD2
CV19/SG/19huG	651–660	IGAEHVNNSY	S1/SD2
CV19/SG/20huG	676–686	TQTNSPRRAR	Furin cleavage site
CV19/SG/21huG	691–699	SIIAYTMSL	S2
CV19/SG/22huG	706–714	AYSNNSIAIP	S2
CV19/SG/23huG	771–775	AVEGD	S2
CV19/SG/24huG	786–789	KQIYK	S2
CV19/SG/25huG	796–800	DFGGF	S2
CV19/SG/26huG	806–820	LPDPSKPSKRSFIED	TMPRSS2 cleavage site and FP1
CV19/SG/27huG	861–866	LPPLL	S2
CV19/SG/28huG	876–890	ALLAGTITSGWTFGA	S2
CV19/SG/29huG	901–910	QMAYRFNGIG	S2
CV19/SG/30huG	920–929	KLIANGFNSA	S2/HR1
CV19/SG/31huG	951–960	VVNQNAQALN	S2/HR1
CV19/SG/32huG	971–980	GAISSVLNDI	S2/HR1
CV19/SG/33huG	996–1105	LITGRLQSLQ	S2
CV19/SG/34huG	1016–1020	AEIRA	S2
CV19/SG/35huG	1046–1055	GYHLMSFPQS	S2
CV19/SG/36huG	1091–1105	REGVFVSNGTHW	S2
CV19/SG/37huG	1111–1115	EPQII	S2
CV19/SG/38huG	1136–1145	TVYDPLQPEL	S2
CV19/SG/39huG	1181–1190	KEIDRLNEVK	HR2
CV19/SG/40huG	1196–1205	SLIDLQELGK	HR2
CV19/SG/41huG	1256–1265	FDEDDSEPVI	CTD

**Table 2 ijms-25-08180-t002:** BLASTp analysis of cross-reactive IgG epitopes for sequence identification to DENV proteins.

Signal (%)	Epitope	aa Position	Sequence	Identity	Serotype	Protein
100	LGVYYHKNNK	141–150	LGVY	75%	DENV2	Polyprotein, RdRp
95.1	LMDLEGKQGNFKNLR	176–190	MDLE	100%	DENV2	Envelope protein
70.5	GAAAYYVGYL	261–270	YVGYL	100%	DENV2	NS5
64.3	GIYQTSNFRV	311–320	NFRV	100%	DENV1	Polyprotein, Helicase
64.3	GIYQTSNFRV	311–320	YQTS	71%	DENV2 and 3	Polyprotein, DEAD domain
60.2	GKIADYNYKL	416–425	GKIA	100%	DENV1 and 2	Envelope protein, partial
60.2	GKIADYNYKL	416–425	KIAD	100%	DENV1	Polyprotein, NS5
63.2	NKKFLPFQQF	556–564	KFLP	100%	DENV2	Polyprotein, NS1
50.3	TQTNSPRRAR	676–686	SPRR	100%	DENV1	Polyprotein, NS1
50.3	TQTNSPRRAR	676–686	PRRA	100%	DENV1, 2 and 3	Polyprotein, NS5 and RdRp
53.3	KQIYK	786–789	QIYK	100%	DENV2	Polyprotein, NS4B
90.1	GYHLMSFPQS	1046–1055	SFPQS	100%	DENV1, 2 and 4	Polyprotein, NS3
90.1	GYHLMSFPQS	1046–1055	MSFP	100%	DENV3	Polyprotein, Envelope protein

## Data Availability

The data presented in this study are available upon request from the corresponding author.

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
