# Peer review of "Enhanced Assessment of Cross-Reactive Antigenic Determinants within the Spike Protein"

_ijms, 2024, doi:10.3390/ijms25158180_

Round 1
Reviewer 1 Report
Comments and Suggestions for Authors
Review IJMS
In this study, the immunodominant IgG B-cell epitopes were mapped. Additionally, they evaluated their cross-reactivity profiles using dengue-positive serum samples. Their results indicated that many epitopes identified in the SARS-CoV spike protein showed similarities with the dengue virus. The authors hypothesized that antibodies against dengue could cross-react with SARS, potentiating SARS infection through antibody-dependent enhancement (ADE).
The authors used 30 serum samples obtained from individuals who received a single dose of a commercial vaccine, plus seven serum samples that received four heterologous doses. Additionally, 45 serum samples positive for dengue and the pre-pandemic period were used. As negative controls, 28 serums were obtained from healthy individuals (blood donors)
Overall, their work as merits and their results could bring new perspectives regarding interactions between host antibodies and SARS infection. Below, I specify the major and specific comments on the manuscript:
Major comment:
In this study, the authors only evaluated cross-reacting antibodies of samples positive for dengue. It would be interesting to test whether serums positive for other wider-spread diseases could also interact. In fact, the authors mentioned in the introduction (lines 65 to 66) that cross-reactivity exists between other endemic viruses and other respiratory viruses. Therefore, it is not entirely clear why they chose dengue only for these analyses.
Minor comments:
By heterologous doses (Line 461). The authors suggested that vaccines are different(?).
The authors measured the LDH release (Figures 4b and 4d). In the discussion, the authors mention that SARS cov-2 trigger pyroptosis, but in this case, the authors did not measure the activation of caspase 1 to determine whether pyroptosis was triggered. For example, lactate dehydrogenase (LDH) is released during necrosis too.
Figure 1 is not clear what rows (H to R) and the columns means. This should be clarified in the revised manuscript. In addition, Figure 1a shows a chemiluminescent image of peptides recognized by antibodies. Perhaps it would be beneficial to organize the dots from darker to lighter. In Figure 1B, the units of the y-axis are not clear. Additionally, I imagine that a higher signal indicated a better interaction between the peptide and antibody (?). If so, it would be beneficial if the information in Table 1 could be organized from higher to lower signals, as in figure 1b, it would be difficult to discriminate between similar signal levels.
Table 1 is an entire page of the manuscript. Perhaps it would be good to consider this table as supplementary material. If the authors/editor consider that this table must go into the manuscript, then it would be good to organize rows from higher to lower signals based on figure 1b.
Figure2a, the y-axis does not have units. Figure 1b shows 41 epitopes, but there are only 24 columns, which is not clear. In addition, the meaning of rows A and B is not clear. Figure1c authors mention that a statistical method was applied to consider significant differences, but in the graph, it is not clear whether all peptides above the 50% threshold are significantly different from those below the 50% threshold or whether there are differences within the group above the threshold.
Table 2 signal lacks unit
There are recent publications that indicate that NSP6 is also a determinant of SARS-CoV2 virulence ( https://www.nature.com/articles/s41586-023-05697-2). It would be interesting to know if any of the BLASTP results run by the authors showed any match against this protein. If so, it would be beneficial to include this in figure 3. In addition, it would be good for the authors to include any comments regarding this in their discussion.
In the materials and methods it is not specified how the authors purified or concentrated the antibodies
Reviewer 2 Report
Comments and Suggestions for Authors
This study identified 41 immunodominant linear B-cell epitopes in the spike glycoprotein using a SPOT synthesis peptide array and serum from hospitalized COVID-19 patients. Bioinformatics revealed a set of epitopes unique to SARS-CoV-2 and potential cross-reactivity with the Dengue virus (DENV). This study underscores the importance of identifying specific epitopes to understand the interplay of past and future infections and their impact on vaccination and immunodiagnostics.
A few concerns:
1. The introduction mentions several SARS-CoV-2 variants of concern (e.g., Alpha, Beta, Gamma, Delta, Omicron XBB.1.5.a) but does not provide specific information on how each variant has impacted transmission, virulence, or vaccine efficacy. Including more detailed data or references on these aspects could strengthen the context.
2. The discussion on antibody-dependent enhancement (ADE) includes references to various viruses but lacks coherence in explaining how these examples relate to SARS-CoV-2. The statement that "no conclusive data has been reported that ADE is related to disease severity" contradicts the earlier emphasis on ADE concerns, creating confusion. More clarity and consistency are needed to present a balanced view of the evidence.
3. The transition between discussing the pandemic's impact, cross-reactivity, and ADE is somewhat abrupt. A clearer connection between these topics would help build a more cohesive narrative. For example, explaining how cross-reactivity and ADE are interconnected in the context of SARS-CoV-2 would improve the flow.
4. The statement that the WHO has declared the end of the COVID-19 pandemic might be misleading or premature, as WHO declarations can evolve, and the situation can change with the emergence of new variants.
5. While the discussion mentions antibody-dependent enhancement (ADE) in the context of both SARS-CoV-2 and DENV, it fails to provide a comprehensive analysis of the in vivo relevance of ADE in COVID-19. The evidence presented is mostly in vitro, which does not always correlate with clinical outcomes.
6. Many points in the discussion heavily cite previous studies without adding new insights or critically evaluating these studies. This reliance can dilute the contribution of the current study to the existing body of knowledge.
Major Revisions
· Provide more specific data on the impact of different SARS-CoV-2 variants on public health and vaccine efficacy.
· Include detailed evidence and references to support claims of cross-reactivity, particularly with DENV.
· Clarify the evidence and arguments regarding ADE, ensuring consistency and coherence.
· Explain the significance of immunodominant epitope mapping and how it contributes to the study's goals.
· Discuss the implications of the findings for vaccine design and public health strategies.
· Update the references to include recent studies that provide current context and support.
· Improve the narrative flow by connecting different sections logically and clearly explaining their relevance to the study’s objectives.
· The discussion needs a more critical analysis, clear linkage between findings and broader implications, and a focus on novel contributions to the field.
Comments on the Quality of English LanguageEnglish needs minor editing.
Round 2
Reviewer 1 Report
Comments and Suggestions for Authors
The authors included sugestions